# Measuring Subaqueous Progradation of the Wax Lake Delta with a Model of Flow Direction Divergence

John B. Shaw[1], Justin D. Estep[1,2], Amanda R. Whaling[1], Kelly M. Sanks[1], Douglas A. Edmonds[3]

[1]Department of Geosciences, University of Arkansas, Fayetteville, 72701, USA
[2]Department of Geology and Geophysics, Texas A&M University, College Station, 77843, USA
[3]Department of Earth and Atmospheric Sciences, Indiana University, Bloomington, 47405, USA

*Correspondence to*: John B. Shaw (shaw84@uark.edu)

**Abstract.** Remotely sensed flow patterns can reveal the location of the subaqueous distal tip of a distributary channel on a prograding river delta. Morphodynamic feedbacks produce distributary channes that become shallower over their final reaches
before the unchannelized foreset slopes basinward. The flow direction field over this morphology tends to diverge and then converge providing a diagnostic signature that can be captured in flow or remote sensing data. Twenty-one measurements from the Wax Lake Delta (WLD) in coastal Louisiana, and 317 measurements from numerically simulated deltas show that the transition from divergence to convergence occurs in a distribution that is centered just downstream of the channel tip, on average 132 m in the case of the WLD. These data validate an inverse model for remotely estimating subaqueous channel tip
location. We apply this model to 33 images of the WLD between its initiation in 1974 and 2016. We find that 6 of the primary channels grew at rates of 60-80 m/yr while the remaining channel grew at 116 m/yr. We also show that the subaqueous delta planform grew at a constant rate (1.72 km$^2$/yr). Subaerial land area initially grew at the same rate but slowed after about 1999. We explain this behaviour as a gradual decoupling of channel tip progradation and island aggradation that may be common in maturing deltas.

## 1 Introduction

River deltas host productive ecosystems and hundreds of millions of people worldwide. Over the past century, river deltas have changed rapidly, putting these large human populations at risk (Barras et al., 2008; Erban et al., 2014; Wilson et al., 2017; Wu et al., 2017). Monitoring morphologic change on river deltas is key to their sustainable management (Peyronnin et al., 2017). Existing remote sensing techniques provide synoptic monitoring of deltas, but are generally limited to monitoring
subaerial or very shallow regions (Couvillion et al., 2011; Li and Damen, 2010; Rahman et al., 2011; Rangoonwala et al., 2016). However, most deltas are far larger than their subaerial portions. For instance, the subaerial area of the Wax Lake Delta in 2015 was 50 km$^2$ (Olliver and Edmonds, 2017) while the subaqueous area was an additional 82 km$^2$ (Shaw et al., 2016a). The difference arises because the Wax Lake Delta has an extensive delta front that lies below low tide. While the subaerial and shallow land area are where marshes are established (Johnson et al., 1985), the subaqueous delta forms the platform upon
which subaerial islands grow (Cahoon et al., 2011; Shaw et al., 2018). Hence, the subaqueous platform extent is important as

a leading indicator of future marsh growth, necessary data for navigation, and the key area metric for estimating delta volume and volume change (Geleynse et al., 2015). Unfortunately, only a small fraction of global river deltas has been directly surveyed in a manner that resolves their subaqueous portion. This is partly due to the vast area of deltas, and partly because year-round turbidity fundamentally limits bathymetric lidar or multispectral remote sensing techniques (Gao, 2009). Many shallow regions along the US coast far from navigation corridors have not been officially surveyed since the 1930s (e.g. NOAA, 2017).

Here we make progress in subaqueous delta monitoring by recognizing key connections between delta front bathymetry and the flow field that organizes over it. We then exploit this connection by remotely sensing the flow direction using streaklines on the water surface visible on some deltas (Figure 1). We use the Wax Lake Delta as a field site due to available bathymetric maps and because it frequently exhibits streaklines that resolve delta front flow directions. In section 2, we review the coupling of emergent delta front bathymetry and flow patterns. In section 3, we present the flow direction to channel (FD2C) model of estimating the location of channel tips using the remotely sensed flow direction field. In section 4, the model is validated on the Wax Lake Delta and with four numerical models of deltas. The model is applied to 33 images of the Wax Lake Delta spanning its development from 1974 to 2016 in section 5 in order to estimate progradation rates of individual channels as well as growth rate of total delta area. The strengths and limitations of the model and results from its application are discussed in section 6.

## 2 Bathymetry and Flow Patterns on River-Dominated Deltas

There is virtually no limit to the paths that a parcel of water can trace across a domain with arbitrary bathymetry and boundary conditions. This seemingly unlimited degree of freedom limits the skill of inverse models predicting bathymetry that place no constraints on the possible morphology (Alpers et al., 2004; Romeiser and Alpers, 1997). However, direct study of river deltas reveals that emergent patterns can be found with their bathymetry and flow patterns. If the bathymetry and flow take on predictable patterns, this greatly reduces the degrees of freedom in a system, improving predictability. The postulation of emergent flow patterns has a long tradition in coastal geomorphology (e.g. Edmonds and Slingerland, 2007; Wright, 1977). Extensive work has been done to predict initial flow and sedimentation patterns associated with turbulent jets entering basins with simple initial bed morphology (Fagherazzi et al., 2015). Our work seeks to extend this approach to systems with complex emergent topography and multiple interacting channels. If a certain bed morphology produces distinct flow patterns visible on remotely sensed imagery, then that pattern can be used to predict the underlying morphology. We make this case for the flow patterns on the delta front of a prograding delta.

For well-developed, prograding river deltas, the bed morphology of a channel terminus can be idealized as an adverse bed slope along the thalweg (shallowing with distance downstream) and a basinward slope along the levee (deepening with distance). Together, these produce a gradual loss of channel confinement (Figure 2). Channels lose definition at the channel tip where the thalweg elevation equals the levee elevation. This transition occurs gradually; over >7 channel widths for the

Wax Lake Delta (Shaw and Mohrig, 2014). Beyond the location where channel definition is lost, the unchannelized delta front grows gradually deeper with distance away from the channel. Although dimensions vary, this general morphology has been observed on the Wax Lake Delta (Shaw et al., 2016b; Shaw and Mohrig, 2014), Brant's Pass crevasse on the birds-foot delta of the Mississippi River (Esposito et al., 2013), the Mobile and Apalachicola river deltas (Edmonds et al., 2011b) and the St.

Clair River Delta (Figure 1b; NOAA, 2017). Additionally, numerical models often produce this morphology (Caldwell and Edmonds, 2014; Geleynse et al., 2010; Liang et al., 2016). These deltas can be qualitatively classified as river-dominated (Galloway, 1975), both by their large fluvial sources and relatively small waves and tides. Waves and tides can alter this morphology (Leonardi et al., 2013; Nardin and Fagherazzi, 2012). Hence, we limit ourselves to river-dominated conditions here.

Recently, flow patterns have been measured across channel tips on the Wax Lake Delta. Various techniques have been used to show that on Gadwall Pass on the Wax Lake Delta (Figure 1a, 3), roughly 50% of water discharge leaves channels laterally (Hiatt and Passalacqua, 2015; Shaw et al., 2016b). This is due to hydrological connectivity between the distributary channels and interdistributary bays across subaqueous levees, and due to reduction of channel cross-sectional area over the final reach of the distributary channel (Coffey and Shaw, 2017; Hiatt and Passalacqua, 2017). One way to track the flow field in this

transitional zone is through streaklines on the water surface. In many coastal settings, slicks of naturally occurring oil and biogenic debris accumulate on the air-water interface (Alpers and Espedal, 2004; Espedal et al., 1996; Garabetian et al., 1993). Despite thicknesses on the order of nanometers, streaks produced by this material are readily observed from boats (Espedal et al., 1996), in near-infrared aerial and satellite imagery as well as from synthetic aperture radar backscatter (Hühnerfuss et al., 1994). Shaw et al. (2016b) showed that the tangent of such streaklines is similar to direct measurements of flow direction, even

when streaklines were mapped and measurements were made months apart. Similar streakline patterns have been observed on other delta fronts as well (Figure 1), and should indicate flow direction where three-dimensional flow patterns and unsteady changes to flow are minimal. Our cursory analysis suggests that streaklines form mostly on deltas with established marshes that are building into freshwater basins or basins where river discharge is enough to make the proximal receiving basin fresh. The latter condition characterizes the Wax Lake Delta and Atchafalaya Bay (Holm and Sasser, 2001; Li et al., 2011).

While streaklines record the depth-averaged flow direction field, they provide no information about the flow velocity magnitude (speed). However, if $\vec{d}$ is the unit vector field aligned with the local flow direction (dimensionless), $h$ is the flow depth field (L), $|\vec{U}|$ is the velocity magnitude field (L/T), and temporal variations ($dh/dt$) are minimal, then conservation of fluid mass can be manipulated to produce a set of equations that relate spatial velocity change ($\breve{A}$), vertical constriction ($\breve{B}$), and lateral divergence ($\breve{D}$; Shaw et al., 2016b):

$$\breve{A} = \breve{B} + \breve{D} \tag{1a}$$

$$\breve{A} = \frac{\nabla |\vec{U}| \cdot \vec{d}}{|\vec{U}|} \tag{1b}$$

$$\breve{B} = -\frac{\nabla h \cdot \vec{\boldsymbol{d}}}{h} \qquad (1c)$$

$$\breve{D} = -\nabla \cdot \vec{\boldsymbol{d}} \qquad (1d)$$

Analysing flow patterns on the delta front downstream of Gadwall Pass on the Wax Lake Delta, Shaw et al. (2016b) found that adverse bed slopes ($\breve{B} > 0$ m$^{-1}$) were generally associated with flow direction divergence ($\breve{D} < 0$ m$^{-1}$). In contrast, downstream of the channel tip on the basinward sloping delta foreset ($\breve{B} < 0$ m$^{-1}$) the flow direction field converged ($\breve{D} > 0$ m$^{-1}$). The transition from negative to positive $\breve{D}$ occurred 400 m (two channel widths) downstream of the channel

tips in that study.

A converging flow direction field for delta front flows is counter-intuitive: turbulent jets emanating from channel mouths generally expand ($\breve{D} < 0$ m$^{-1}$) with distance downstream (Kundu et al., 2011) due to lateral shear with still water or bed friction. Qualitatively, jets expand because deceleration promotes an increase of cross-sectional area of the jet core. When the flow depth is constant or decreasing, this increase in area is accomplished by widening and flow direction divergence.

However, when the depth increases rapidly compared to the increasing cross-sectional area (and the jet does not detach from the bed), then the jet width must contract and flow directions must converge. A scaling of shallow water jets by Özsoy and Ünlüata (1982) showed that jets can converge or contract in width when the basinward bed slope ($\nabla h \cdot \vec{\boldsymbol{d}} > 0$) exceeds the dimensionless Darcy-Weisbach friction factor divided by 8 (equivalent to the commonly used friction factor $C_f$). Recent numerical modelling by Jiménez-Robles et al. (2016) also shows that jets can exhibit flow direction convergence when

basinward slopes exceed ~1%. The maximum foreset slopes on the Wax Lake Delta are about 2 x 10$^{-3}$, which is slightly too gradual to produce flow direction convergence from either of these studies. However, we note that there is a physical basis for flow direction convergence on delta fronts that supports the convergence we observe in streaklines.

## 3 The FD2C Model

If a channel tip's location controls the flow direction field, we seek an inverse method of estimating the channel tip location

from the flow direction field that can be used with remotely sensed imagery. Previous analysis (see Section 2) showed that the transition from adverse bed slopes to basinward bed slopes at channel tips is coupled with the transition from flow divergence to flow convergence as tracked by streaklines. We name this model of coupled bathymetry and flow the C2FD model ("Channel to Flow Divergence"), where channel tips control the flow direction field. If this correlation is persistent and predictable, then the location of the channel tips can be related to a critical point in the $\breve{D}$ field ($x_{\breve{D}}$). Analysis of the Wax Lake Delta and

numerical delta simulations show that $x_{\breve{D}}$ is where $\breve{D} = 0$ m$^{-1}$ and $\breve{D}$ is changing from negative to positive in the downstream direction (Figure 2). We name this inverse model FD2C for "Flow Divergence to Channel tips."

## 3.1 Application

The $\breve{D}$ field can be calculated using streaklines (Figures 3, 4). First, we trace the curvilinear shape of all streaklines manually in ArcGIS. Streaklines are also mapped down the center of primary distributary channels if there is a streakline or not. This is done because flow direction is generally found to follow the trends of large channels. Assuming that the local flow direction is everywhere tangent to the streakline, we sample each streakline at 25 m increments along the line, noting the local direction of the line. This produces a dataset of points $P(x', y', \vec{d})$, where $x$ and $y$ are the Easting and Northing spatial coordinates (UTM Zone 15N) and $\vec{d}$ is the unit vector tangent to the mapped streakline. Flow direction $\vec{d}$ is recorded as a unit vector with components in the $x'$ and $y'$ directions, which for which we use Easting and Northing coordinates: $\vec{d} = (d_{x'}, d_{y'})$, $(d_{x'}^2 + d_{y'}^2 = 1)$. The flow direction field is then constructed by interpolating $d_{x'}$ and $d_{y'}$ independently from $P$. We use the biharmonic spline interpolation technique of Sandwell (1987) because of the smooth interpolation results. The resulting fields were again normalized by their magnitude to insure the field remained unit vectors. Finally, the flow convergence field $\breve{D}$ is calculated on the grid as $\breve{D} = -\nabla \cdot \vec{d} = -\left(\frac{\partial d_{x'}}{\partial x} + \frac{\partial d_{y'}}{\partial y}\right)$ (Figure 3b). For numerical models (Figure 4), the $\breve{D}$ field was calculated directly from the modeled depth averaged velocity field and thus required no interpretation of streaklines or interpolation, so the $\breve{D}$ field is exact in that case.

## 3.2 Estimating Channel Tip Location

We test the FD2C model by comparing the location of channel tips to the critical divergence point $x_{\breve{D}}$ on the Wax Lake Delta, as well as on a set of numerically modelled deltas. For each primary distributary channel, we draw a transect down the center of the subaerial reach of the distributary channel extending into the basin (Figure 4) and track bathymetry ($\eta(x)$) and flow direction divergence ($\breve{D}(x)$) as a function distance $x$ along it (Figure 5). The channel tip, $\hat{\eta}$ is defined as the global maximum elevation along the transect and the location defined as $x_{\hat{\eta}}$. The critical divergence point ($x_{\breve{D}}$) is defined as the first downstream location where both $\breve{D} = 0$ m$^{-1}$ and $d\breve{D}(x)/dx > 0$ m$^{-2}$. Note that this location is interpolated along $\breve{D}(x)$, and therefore may appear to have sub-grid resolution. The difference $\Delta l = x_{\breve{D}} - x_{\hat{\eta}}$ is defined as the distance downstream of the channel tip where the critical divergence point occurs (Figure 5). Note that if $\Delta l < 0$ m, then the critical divergence point occurs upstream of the channel tip.

The method is designed to estimate one channel tip location that is along the subaqueously defined distributary channel axis. The benefit is that this channel axis is easily defined in imagery, but it means that the method cannot account for bends or branches in the subaqueous reach.

### 3.3 Validation

Summary statistics of $\Delta l$ (Table 1, Figure 6) provide a means of testing the FD2C model, which indicates that $\Delta l$ is generally small. Data was drawn from the Wax Lake Delta by comparing delta front bathymetry collected in July 2010 (2 channel tips), August 2011 (6 tips), February 2015 (6 tips), and July 2016 (7 tips) to imagery from 14 October 2010, 1 October 2011, 19

April 2015, and 5 April 2016 respectively. Over these 21 measurements, $\Delta l$ had a mean of 145 m and a median of 132 m (Figure 6a). The sample had an interquartile range of 701 m (Table 1, Figure 6). This supports the claim that $\breve{D}_{cr}$ is generally near $\hat{\eta}$, but also shows that the variance is large.

The distribution of $\Delta l$ on the Wax Lake Delta was compared to the unsteady hydrologic conditions present when the aerial image was collected (Figure 7; data are in supplementary material). The measurements were made over river discharge values

spanning low flow to flood discharge upstream of the delta at Calumet, LA, (USGS #07381590). Measurements also spanned very low tide to relatively high tide, and characteristic rates of rising and falling tide measured at Amerada Pass, in Atchafalaya Bay (NOAA #11354). In each case, the variation in $\Delta l$ was far larger than any correlation with these parameters, and $r^2$ values were each less than 0.02. Even if the linear fits were statistically significant, they would explain no more than 200 m variation in $\Delta l$ over the common values of discharge and tidal conditions. These analyses suggest that unsteady flow is not an important

control on the distribution of $\Delta l$.

In order to achieve some validation independent of the Wax Lake Delta, the FD2C model was also evaluated on four numerical river deltas originally presented by Caldwell and Edmonds (2014). These deltas were modelled using Delft3D on a 25 x 25 m$^2$ grid. Model runs A1a1, A1e1, D1a1, and D1e1 were used. These runs had an upstream discharge of 1000 m$^3$/s and no tidal or wave forcing. They differed in incoming median grain diameter between 0.01 and 0.1 mm, the sorting of the sediment

distribution, and the fraction of the sediment that was cohesive. Full descriptions of the runs are found in Caldwell and Edmonds (2014). Measurements began at time step 500 to allow a significant deposit to develop and then every 5 time steps thereafter. At each time step, up to 5 of the largest distributary channels in terms of flow velocity were measured. Fewer measurements were made if less than 5 channels were present. These analyses yielded a total of 374 samples (Table 1).

For the four modelled deltas, the median $\Delta l$ ranged from 12 m to 199 m (Figure 6a). While the ranges of $\Delta l$ were up to 2110

25    m in the case of D1e1, the interquartile ranges were between 156 to 254 m. Some transects drawn on numerical deltas did not yield $x_{\breve{D}}$ because the criteria for $x_{\breve{D}}$ were not met. In these failed cases, the $\breve{D}$ transect was always positive, or trended from positive to negative. This meant that $\Delta l$ could not be measured and the FD2C method could not be applied. Such cases accounted for 17% of the transects on delta A1a1 and 21% of delta D1a1 (Table 2) and 8% of the total transects measured on numerical deltas.

Measurements from the modelled deltas also show the distribution of $\Delta l$ is relatively stable over time (Fig. 6b). A linear regression was fit to $\Delta l$ versus time and the slope of the data was not significant by a *t*-test ($p > 0.10$ for each numerical delta). The slope that was found ($1.6 \pm 2.6$ m/yr for D1e1) would introduce a small error to $\Delta l$ relative to the uncertainty of $\Delta l$ (order 100 m) even if it slowly grew over many decades. This near-stationarity suggests that $\Delta l$ can be assumed constant in time,

even as a delta progrades. We also investigated whether $\Delta l$ is a function of upstream channel width and flow depth at the channel tip (Supplementary Material). However, none of these parameters showed predictive power over $\Delta l$.

Taken together, these analyses validate the FD2C method for prograding deltas with several distributary channels. Measurements from the Wax Lake Delta and numerical models all show that the central tendency is for $\Delta l$ to be about 100 m with little dependence on unsteady hydrologic conditions, and model data shows that the distribution remains stationary over time. We apply this result to the Wax Lake Delta to measure the growth of its subaqueous channel tips.

## 4 Tracking Wax Lake Delta progradation with the FD2C model

### 4.1 Methods

The FD2C method was applied to estimate the locations of channel tips over time on the Wax Lake Delta using 33 images between 30 January 1974 and 5 April 2016. Images are near infrared imagery from Landsat 2, 5, and 8, SPOT and an overhead photomosaic. See Supplementary Material for imagery metadata and Acknowledgements for image availability. For each image, the FD2C method (Section 3) was applied by mapping a transect starting at the edge of subaerial exposure (delta shoreline) and extending along the 7 primary distributary channel axes of the WLD (Figure 3b) to find $x_{\breve{D}}$. The first two images (30 January 1974 and 9 February 1979) showed minimal subaerial delta exposure, so transects were mapped over abrupt changes in $\breve{D}$ and grouped to East Pass in the eastern portion of the delta, Gadwall Pass in the central portion of the delta, and Campground Pass in the western portion of the delta. The channel tip location was then estimated as $x_{\breve{D}} - \Delta l$, or an average 145 m upstream of $x_{\breve{D}}$ according to measurements from the Wax Lake Delta itself (Figure 6, Table 1). Channel tip growth was then tracked as the Euclidian distance between the delta apex (UTM Zone 15N: 651673 E 3267186 N).

Estimated channel tips were connected to one another and the pre-delta shoreline to measure the area within the delta's subaqueous platform. Each channel tip occurs at a crest in bathymetric elevation and progrades via erosion of the deposit in front of it. By connecting these tips, we enclose an area that has received significant deposition, but not yet enough to become subaerially emergent, even at low tide. The enclosed area also contains channels and all subaerially emergent regions, but excludes some sea-ward deposition associated with the delta forest. We name the region the total delta area. Monte-Carlo techniques were used to include uncertainty in $\Delta l$ in calculating this area. First, $x_{\breve{D}}$ was found for each of the seven primary distributary channels using the technique above. The location of $x_{\hat{\eta}}$ was determined by randomly sampling (with replacement) one of the 21 measured values of $\Delta l$ that were measured on the Wax Lake Delta (Figure 6a), and then estimating the location of the channel tip $\left( x_{\hat{\eta}} = x_{\breve{D}} - \Delta l \right)$. The seven channel tips were then connected by straight lines, and then connected to a pre-delta shoreline of Atchafalaya Bay mapped from 1974 imagery (Figure 3). The pre-delta shoreline extends 10 km up the original Wax Lake estuary (Shlemon, 1972), however delta area is truncated north of 3269274 N in order for area results to be more comparable to existing datasets. The truncated area of the original Wax Lake estuary is 17.2 km$^2$, and can be added to all area estimates if desired. In order to join the straight lines connecting channel tips to the pre-delta shoreline, the East Pass

channel tip was connected to the pre-delta shoreline with a ray with a 27˚ azimuth (Figure 3b). The Campground Pass channel tip was connected to the mainland with a ray of 0˚ azimuth. These azimuths were chosen to accurately reflect the marginal deposition on the Wax Lake Delta over the imagery used in the study. Total area is not sensitive to these choices. The area of the resulting polygon was calculated $10^4$ times with different random sampling to account for the distribution of $\Delta l$ (Monte-Carlo sampling). The 16th, 50th, and 84th percentile of areas were then recorded for a given image. This process was repeated for each image to track delta area over time.

## 4.2 Results

Channel tip progradation rates are shown in Fig. 8. Between 1974 and 2016, each of the seven primary distributary channels extended at least 2 km. In clockwise order, East, Pintail, Greg, Main, Gadwall, Mallard, and Campground Passes had average progradation rates of 74 ± 9 m/yr, 75 ± 13 m/yr, 89 ± 13 m/yr, 73 ± 10 m/yr, 116 ± 10 m/yr, 66 ± 15 m/yr, 60 ± 20 m/yr, respectively. All primary distributary channels except Gadwall Pass grew at rates between $60 \pm 20$ (Campground) and $89 \pm 13$ m/yr, which are nearly indistinguishable given the uncertainty. In contrast, Gadwall Pass grew at a significantly faster rate of $116 \pm 10$ m/yr. Looking beyond simple linear regression, we used the "segmented" package in R (Muggeo, 2003) to search for break-points, or dates with different progradation rates before and after. However, no statistically significant break-points were found.

The delta area estimated using the FD2C model is shown in Fig. 9. The delta area shows an apparently linear increase in area over time from 38.6 km$^2$ in 1974 to 113.4 km$^2$ in March 2016. The growth rate over this period is 1.72 ± 0.13 km$^2$/yr, and maintains this trend remarkably well over decadal timescales. The data have a root-mean-square error of about 7.86 km$^2$ associated with the Monte-Carlo sampling of area. This uncertainty is generally smaller than the residuals where datapoints departed from the linear trend, which averaged 13.69 km$^2$ over the dataset.

## 5 Discussion

### 5.1 The FD2C Model

The FD2C conceptual model assumes that water leaving a self-formed distributary channel will have a flow direction field that first diverges $(\breve{D} < 0)$ and then converges $(\breve{D} > 0)$ with the transition between the two fields occurring near the channel tip where the bathymetric elevation peaks and begins to slope basinward. This model was supported by measurements from Wax Lake Delta and four Delft3D model runs (Figures 5, 6). Analysis of $\Delta l$ using modelled deltas also confirms that temporal trends in $\Delta l$ are insignificant. We use this to assume that the distribution of $\Delta l$ is stationary and the modern distribution can be applied to the delta in the past. This is important, because field measurements on the Wax Lake Delta from the past 5 years do not provide the time span to confirm this in a field setting.

Each of the $\Delta l$ distributions have significant standard deviations (Table 1). This suggests that although the FD2C conceptual model is accurate to first order, other processes also affect $\Delta l$. Analysis of unsteady conditions showed that they had little predictive power over $\Delta l$ measured on Wax Lake Delta (Figure 7). Wind shear could also play a role, but local wind measurements were only available for two of the four images where $\Delta l$ was measured, and in each case the wind was light (<4 m/s) and from the east. Hence, we have insufficient data to test the effect of wind setup at this time. However, the wide spread of $\Delta l$ even for a single image makes wind control unlikely.

There is also uncertainty associated with the use and interpolation of the streaklines in calculating the $\breve{D}$ field in field cases. Analysis of $\Delta l$ on the Delft3D deltas showed standard deviations that were 35-81% of the standard deviation on Wax Lake, suggesting that even when unsteadiness and interpolation errors are neglected, the variation of $\Delta l$ remains significant. Given that unsteadiness and interpolation errors explain only portions of the $\Delta l$ distribution, we hypothesize that the remainder stems from channel properties. If two channels are near one another, their outflows and the unchannelized flow between them would be constricted compared to a channel that is far from its neighbours. This may be particularly important in places like the Eastern portion of the Wax Lake Delta, where Main, Greg, Pintail, and East passes enter Atchafalaya Bay over about 8 km (Figure 3a). Channel dimensions and input discharges also affect hydrodynamics including the $\breve{D}$ field. Finally, aspects of channel tips that have been less studied, such as the bed slope of the channels or number of branches in the subaqueous delta front, could have important effects. We expect that further study of flow patterns and bed morphology on complex deltas will shed more light on what sets $\Delta l$.

The variation of $\Delta l$ prevents confident interpretations of changes in channel tip location on seasonal or annual timescales. For example, extension and back-stepping of channel tip location on the order of several hundred meters were directly measured on Gadwall Pass between July 2010 and February 2012 (Shaw and Mohrig, 2014). The 512 m standard deviation of $\Delta l$ prevent these changes from being estimated with confidence. Even with this limitation, certainty in measuring change increases with time. The ~60-116 m/yr growth rates observed on distributary channels grow larger than the standard deviation after 4-9 years. For platform area growth analysis, the standard deviation produced by Monte-Carlo sampling of $\Delta l$ measurements is about 8.2 $km^2$, and progradation rates are calculated to be 1.72 $km^2$/yr. Hence, for estimating changes in Wax Lake Delta total area, we expect the method to be able to perform on timescales of greater than 4-5 years. Clearly, more research on the effects of channel and delta morphological characteristics and unsteady flows on $\Delta l$ are warranted, and could increase the sensitivity of the FD2C method for monitoring change. However, the method already detects clear changes in channel tip and delta area at the decadal scale or better for the Wax Lake Delta.

Where can the FD2C model be applied? The model was validated on deltas that were prograding with several active channels where lateral channel migration was minimal, and there are many tens of deltas globally that have these characteristics. Streaklines must also be present if the $\breve{D}$ field is to be estimated from remote sensing, and streaklines are somewhat rarer, but our cursory search has revealed at least 10 deltas globally (e.g. Fig. 1) with detectable streaklines under certain conditions. In particular, streaklines are common around the bird's foot delta of the Mississippi River's main stem where many coastal restoration projects are planned or are currently operational. One example is the West Bay Diversion, where the progradation

of a delta is an explicit goal (Allison et al., 2017; Andrus and Bentley, 2007; Kolker et al., 2012). The method can also be used with the decades of remote sensing imagery that already exist. Our analysis of the Wax Lake Delta found that streaklines were sometimes visible in Synthetic Aperture Radar backscatter, Landsat 1, and CORONA imagery, making monitoring from the 1960s or earlier potentially feasible. The FD2C model could also be used as a hypothesis for the hydrodynamic study of other

similar deltaic systems, even if they do not support streaklines. For example, flow direction convergence occurred on nearly all delta foresets analysed here, so it is possible that this pattern occurs but remains unmeasured on many of the world's river deltas. Converging flow patterns could inform the study of delta front sedimentology (e.g. Enge et al., 2010) or the hydrodynamics of a plunging river plume (Lamb et al., 2010). Whether applied to monitor growth or understand delta front hydrodynamics, we present the FD2C model as an advance in understanding the coupled morphology and flow field at

distributary channel tips.

### 5.2 Subaqueous growth of the Wax Lake Delta

### 5.2.1 Channel Tip Progradation

The FD2C model allows the progradation rates of individual subaqueous channel tips on the Wax Lake Delta to be measured, and provides new insight into decadal growth patterns of the Wax Lake Delta from its initiation to present. The hypothesis of

radially symmetric growth often applied to the Wax Lake Delta (Kim et al., 2009; Paola et al., 2011) is largely supported: six of the seven channels have prograded at rates between 60 and 90 m/yr. However, the consistently larger progradation rate of Gadwall Pass ($116 \pm 10$ m/yr) also suggests that the delta is becoming more asymmetric over time. Future evolution may correct for the dominance of Gadwall Pass, possibly by a soft avulsion (*sensu* Edmonds et al., 2011a) where Gadwall Pass's progradation decelerates and another channel accelerates. However, the consistently dominant growth rates since 1983 (Figure

8) and the fact that Gadwall Pass is presently the widest channel (Figure 1a, 3a) suggests that this dominance will continue and delta asymmetry will continue to grow.

### 5.2.2 Delta Area Growth

Previous studies of delta growth have focused on the emergence of subaerial land using Landsat imagery. Allen et al. (2012) investigated the area of subaerial land growth over the entire Wax Lake Delta. They determined a growth rate in Landsat

imagery as a function of time, water discharge, and tide level. They found that the subaerial delta grew at a rate of 1.1 km$^2$/yr between 1983 and 2002, and then reduced growth to near zero afterward. Olliver and Edmonds (2017) focused on emergence of just the central islands of the WLD, neglecting some marginal areas of the delta included by Allen et al. (2012). Analyses were based on two images per year selected for minimum and maximum biomass, which mitigated the large swings in area shown by Allen et al. (2012). The authors also interpreted a break in growth rate at about 1999, with a growth rate from 1984-

1999 of $1.88 \pm 0.42$ km$^2$/yr and a growth rate from 1999-2015 of $0.78 \pm 0.44$ km$^2$/yr. The total delta area as measured by the FD2C method grew at a rate of $1.72 \pm 0.13$ km$^2$/yr from 1974 to 2016 without any break-points in growth rate.

In all cases, the FD2C method produces delta area estimates that are >40 km$^2$ larger than estimates of sub-aerial land. The FD2C method is designed to track the location of subaqueous channel tips significantly below any water level datum. Furthermore, the FD2C method includes distributary channels and subaerial land as part of the delta area. Therefore, it stands to reason that the area estimates would be far larger than subaerial methods. However, we also note that the total delta area as it is defined here neglects some foreset deposition, so is not necessarily an upper bound on delta area. The focus on the Wax Lake Delta also ignores the ~10,000 km$^2$ deposit of silt and clay accumulating on the Atchafalaya Shelf that is fed by the Wax Lake and Atchafalaya Deltas (Draut et al., 2005; Neill and Allison, 2005).

Despite the larger absolute area, the FD2C method yielded long-term growth rates that were broadly similar to subaerial growth rates from initial emergence until 1999 (Figure 9). During that time, the subaqueous platform was always ~40 km$^2$ larger than the subaerial platform, suggesting that the subaqueous platform was being converted to intertidal or subaerial land at a rate similar to the rate of subaqueous platform production through progradation.

From 1999 to present, the subaqueous total delta area creation continued unabated, but the processes converting subaqueous platform to intertidal or subaerial land reduced by 59% or more. We suggest that this is a consequence of channels prograding radially, and becoming further apart with distance from the delta apex. Field and remote sensing studies of the Wax Lake Delta have confirmed that island aggradation rates decrease and timescales of emergence increase with distance from the edges of the primary channels (Bevington and Twilley, 2018; Olliver and Edmonds, 2017; Wagner et al., 2017). We reason that increased island width limits vertical accretion because suspended sediment from the channels must travel farther through a vegetated island. Despite this reduction in subaerial growth rate, the total delta area growth rate was roughly constant, consistent with a roughly constant sediment supply. The growing disconnect between subaqueous progradation and island aggradation suggests that these processes begin to decouple as deltas become more mature.

This has interesting implications for the Wax Lake Delta and coastal restoration initiatives. The ecology (Carle et al., 2015; Olliver and Edmonds, 2017) and carbon sequestration (Shields et al., 2017) are highly dependent on island elevation, and the ratio of subaerial land to total delta area appears to decrease time. The coupled subaerial-subaqueous monitoring scheme used to discover this transition is likely transferrable to other large-scale coastal restoration efforts in Louisiana (streaklines are frequently observed across coastal Louisiana), allowing sediment accumulation and marsh formation to be independently tracked.

## 6 Conclusion

The morphodynamic evolution of channel mouths can produce flow patterns and bed morphology that are closely coupled. The delta front morphology consists of subaqueous channels that grow shallower in the downstream direction, subaqueous levees that allow water to exit the channel laterally, and a sloping delta foreset. The flow direction field over this morphology diverges in the final reach of the channel and converges on the delta foreset. The location of the transition from divergence to convergence relative to the channel tip ($\Delta l$) varies by many hundreds of meters, but is on average 0-200 m downstream of the

channel tip in both field data from the Wax Lake Delta and numerically modelled deltas. This distribution of $\Delta l$ appears to be independent of hydrodynamic unsteadiness. It also appears stationary, allowing it to be applied through time. We present the FD2C method to relate the divergence of flow direction estimated using remote sensing of streaklines on the water surface to channel tip location with quantitative uncertainty.

The FD2C method provides a means of estimating the progradation of channel tips and total delta area of the Wax Lake Delta from its initiation in 1974 through 2016. The method involves uncertainties associated with flow field characterization and channel tip location estimation. However, the method allows key aspects of the Wax Lake Delta's progradation to be characterized for the first time such as individual subaqueous channel tip progradation rates and the growth of the total delta area. Channel tips grow at rates ranging from 69-116 m/yr that appeared constant at the decadal scale. The subaqueous total

delta area also grew steadily between 1974 and 2016 at a rate of $1.72 \pm 0.13$ km$^2$/yr. The reduction of subaerial growth rates below this rate around 1999 suggest that the Wax Lake Delta has become less efficient at building subaerial land as islands have grown wider, even as subaqueous deposition continued unabated. This monitoring techniques furthers our understanding of the Wax Lake Delta, which is a example of an uncontrolled river diversion being investigated as a possible land building strategy in Louisiana. The FD2C model can be applied to similar deltas where direct field measurements are impossible or

scarce.

## Appendix A. Notation

| | |
|---|---|
| $\breve{A}$ | Fractional Velocity Increase in the downstream direction (m$^{-1}$) |
| $\breve{B}$ | Fractional Bed constriction in the downstream direction (m$^{-1}$) |
| 20    $\breve{D}$ | Divergence in flow direction (m$^{-1}$) |
| $\vec{d}$ | Unit vector aligned with flow direction (dimensionless) |
| $\Delta l$ | distance downstream of the channel tip where the critical divergence point occurs; $\Delta l = x_{\breve{D}} - x_{\hat{\eta}}$ (m) |
| $\lvert \vec{U} \rvert$ | Velocity magnitude (m/s) |
| $x_{\breve{D}}$ | Critical divergence point along an axial channel transect (m) |
| 25    $x_{\hat{\eta}}$ | Elevation crest along an axial channel transect (m) |
| $x'$ | Easting coordinate in Universal Transverse Mercator (UTM) reference frame (m) |
| $y'$ | Northing coordinate in Universal Transverse Mercator (UTM) reference frame (m) |
| $\eta$ | Bed elevation (m) |

30

**Author Contributions**

J.B.S. conceived and led the study. J.D.E. did the first streakline mapping and analysis and proposed the criteria for $x_{\tilde{D}}$. A.R.W. performed the 2016 bathymetric survey. K.M.S. contributed the statistical analyses of break-points. D.A.E. provided the numerical model simulations and Landsat imagery. J.B.S. wrote the manuscript with contributions from all co-authors.

## 7 Acknowledgements

Landsat imagery used in this study were downloaded from Google Earth Engine. Bathymetric maps of Wax Lake Delta are available in (Shaw, 2013; Shaw et al., 2016a). Imagery metadata, channel tip location estimates, and timeseries of area are all available in the supplementary material of this paper. Interpolated images are available in an online repository (Shaw & Haynes, 2018). This work was supported by a U.S. Department of Energy grant to J.B.S. (DESC0016163). Kathryn Hurlbut performed the measurement of the numerical deltas and was involved in the early writing of this study. Ashlyn Haynes contributed valuable work mapping many of the Landsat images for streaklines. Julie M. Cains illustrated Fig. 2. We thank Elizabeth Olliver for providing data from her study, Brad Murray for helpful criticism, and two anonymous reviewers for constructive criticism that led to an improved manuscript.

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

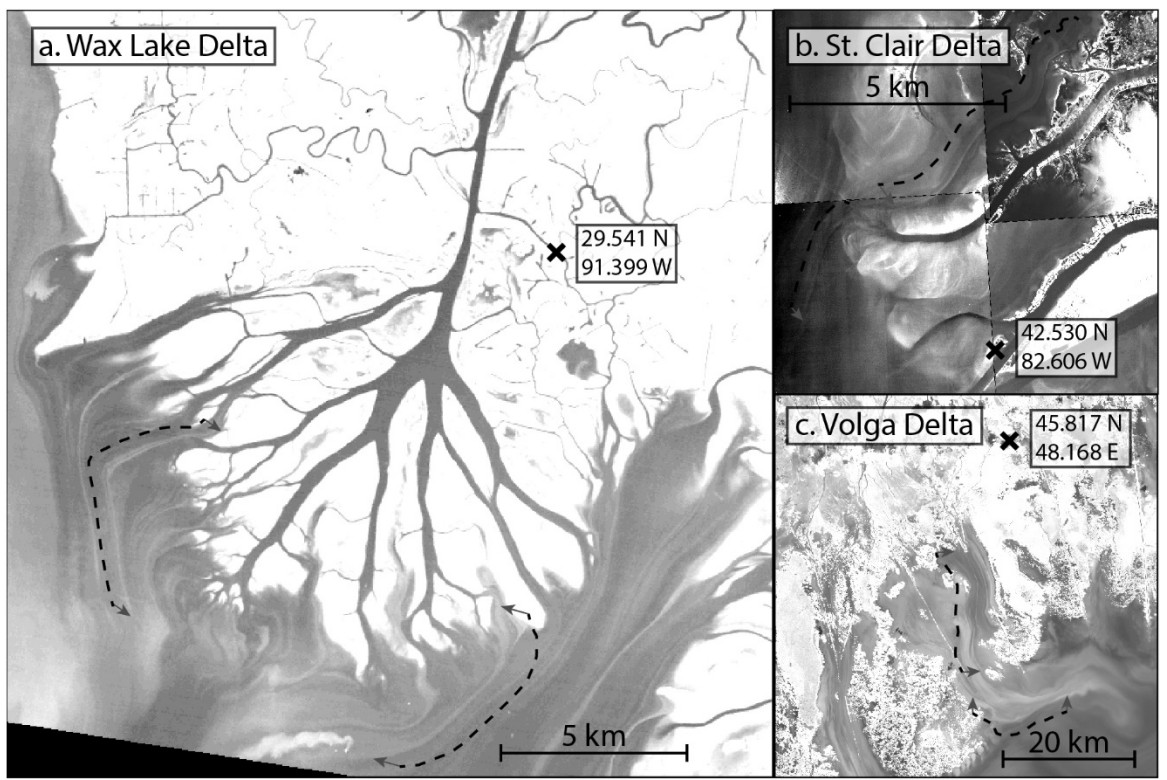

**Figure 1: Images of river deltas exhibiting streaklines. (a) Wax Lake Delta (Landsat image LT50230402011002CHM01 Band 4). (b) the Saint Clair Delta in Michigan, USA (Digital Orthophoto Quads Saint Clair Flats NE, NW, SW, SE, IR band). (c) Portion of the Volga Delta in Russia (Landsat LC08_L1TP_168028_20170501_01 Band 4). In each image, streaklines are mapped in dashed lines. The line is translated in space slightly (see gray arrows) so as not to cover the streakline in the image.**

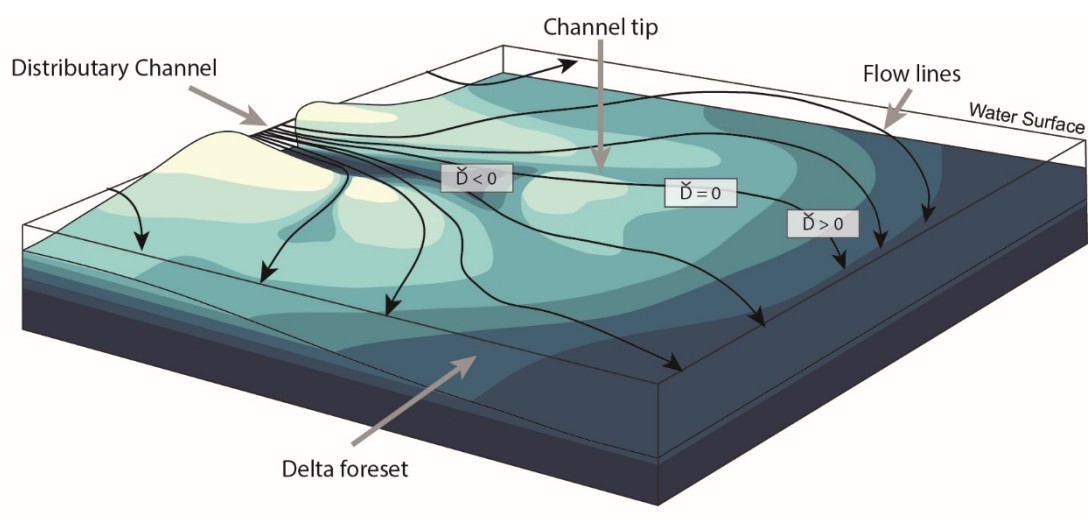

**Figure 2: Schematic diagram of delta front morphology and streakline behavior. The colormap shows topography with dark colors representing deep areas and light colors representing shallow or subaerial areas. Streaklines are shown as black solid lines. The FD2C method takes advantage of flow direction divergence $(\breve{D} < 0)$ through the shoaling reach of the channel and lateral flow direction convergence $(\breve{D} > 0)$ on the basinward sloping delta front. The channel tip occurs roughly wher $\breve{D}$ transitions from negative to positive.**

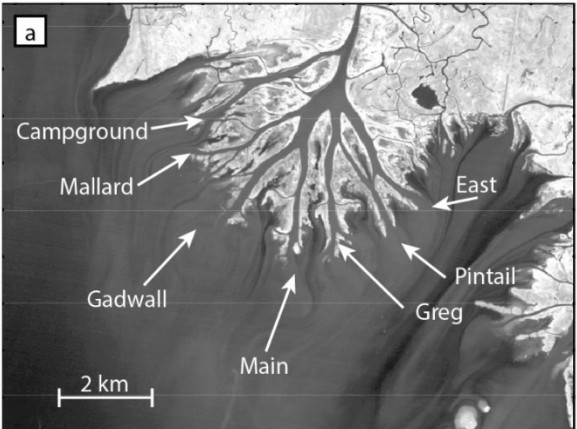

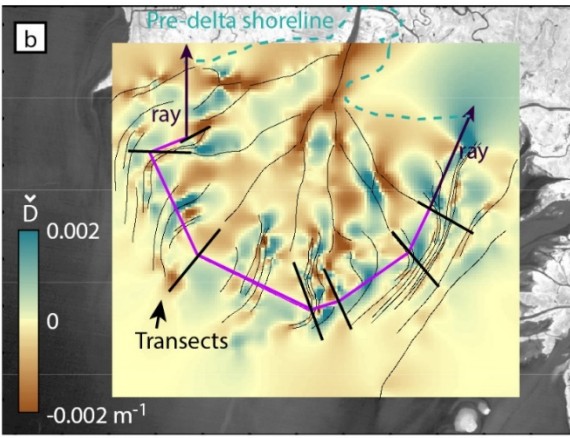

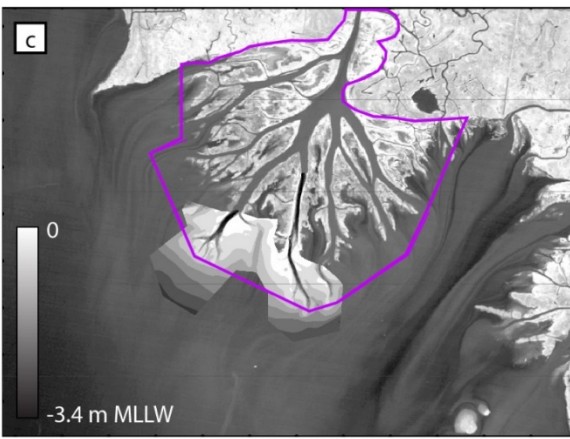

**Figure 3. Method for converting imagery into channel tips and delta area. (a) Landsat image displaying streaklines (14 October 2010; Supplementary Material). The seven primary distributary channels are labelled. (b) Streaklines (thin black lines) are mapped manually on the delta front, and lines are also placed down the center of subaerially emergent distributary channels. The $\check{D}$ field is interpolated from these streaklines (colormap). Thick black lines are transects extending from the seven primary distributary channels. The estimated location $x_{\hat{\eta}}$ along each transect is connected via the purple line and rays connect channel tips to the pre-delta shoreline to close the area. (c) The interpreted total delta area is shown. A bathymetric map from June 2010 referenced to mean lower low water (MLLW) shows how the interpreted channel tips compare to direct measurements.**

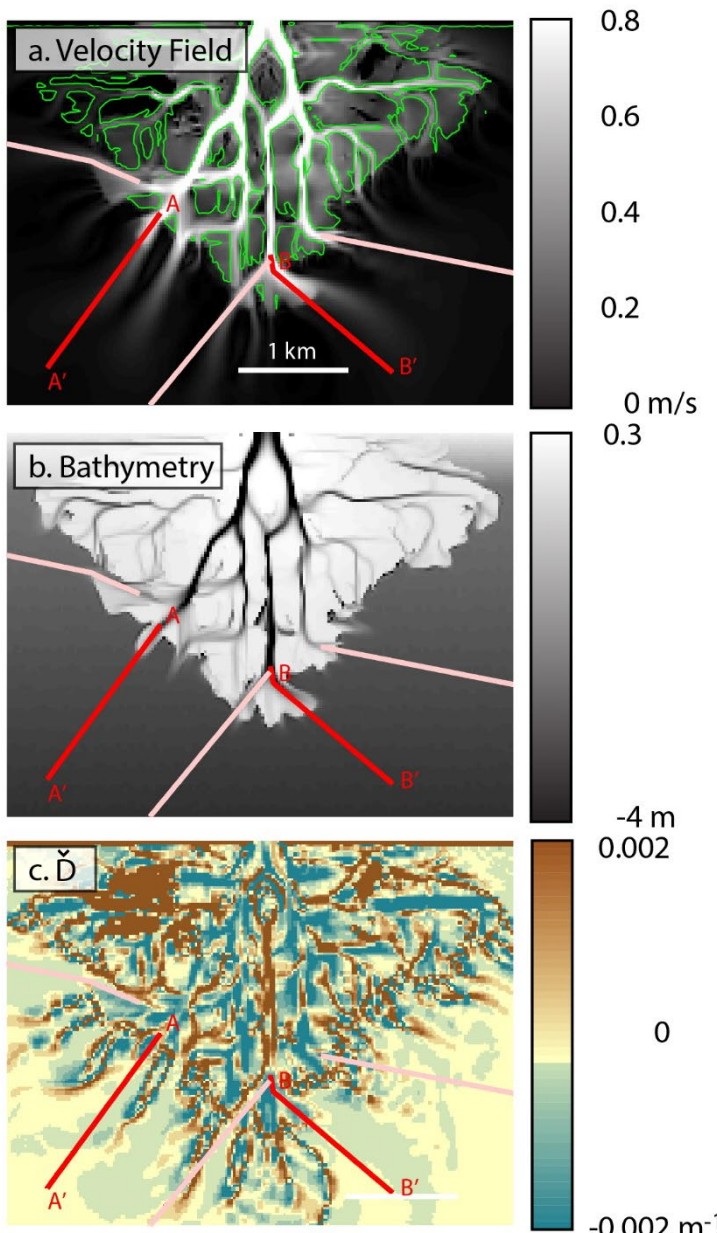

**Figure 4. Method for comparing $\breve{D}$ to bathymetry using a Delft3D numerical simulation (run A1e1). (a) The velocity field and -0.3 m MSL (green) contour are displayed and transects (pink and red lines) are drawn extending from the largest distributary network channels. (b) The bathymetric profile is collected along each transect. (c) $\breve{D}$ is calculated, and transects of $\breve{D}$ are collected along the transects. Transects A-A' and B-B' are shown in Figure 5.**

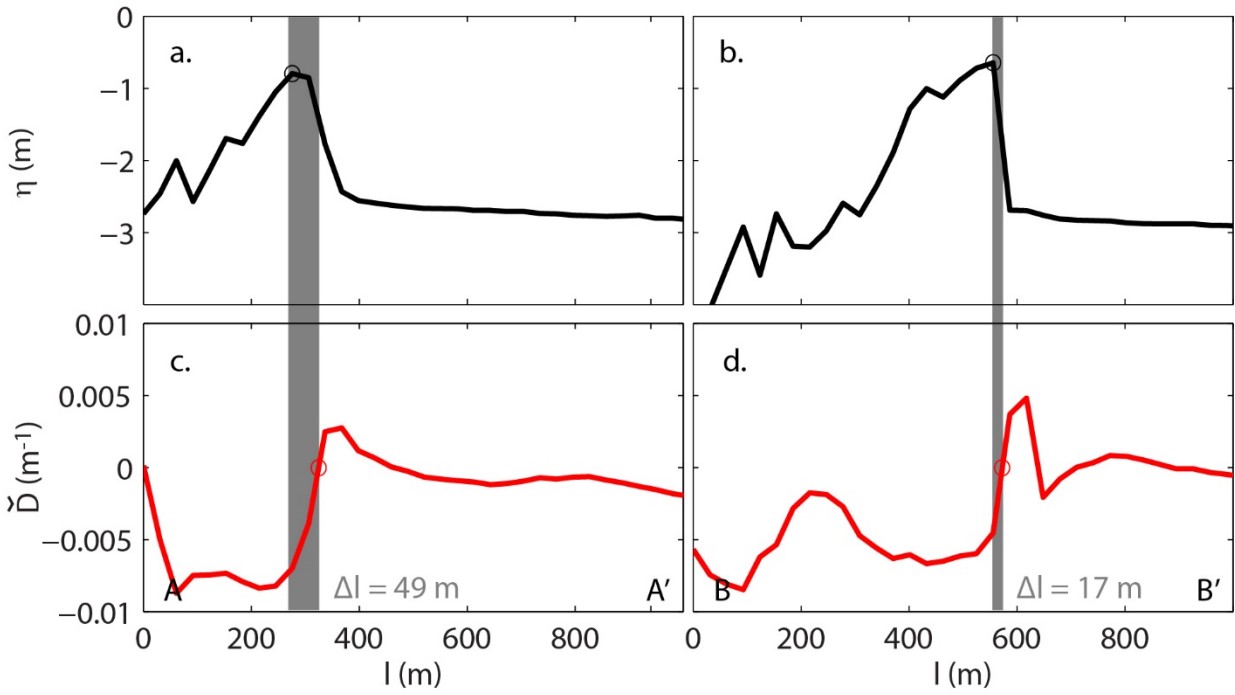

**Figure 5. Comparison of bathymetry (η; plots a and b) and divergence of flow direction ($\breve{D}$; plots c and d) for transects A-A' (plots a, c) and B-B' (plots b, d). Δ*l* is the width of the gray box, or the location where $\breve{D}$ changes from positive to negative ($x_{\breve{D}}$; red circle) minus the bathymetric maximum of the channel tip ($x_{\hat{\eta}}$; black circle). The distribution of Δ*l* is shown in Figure 6.**

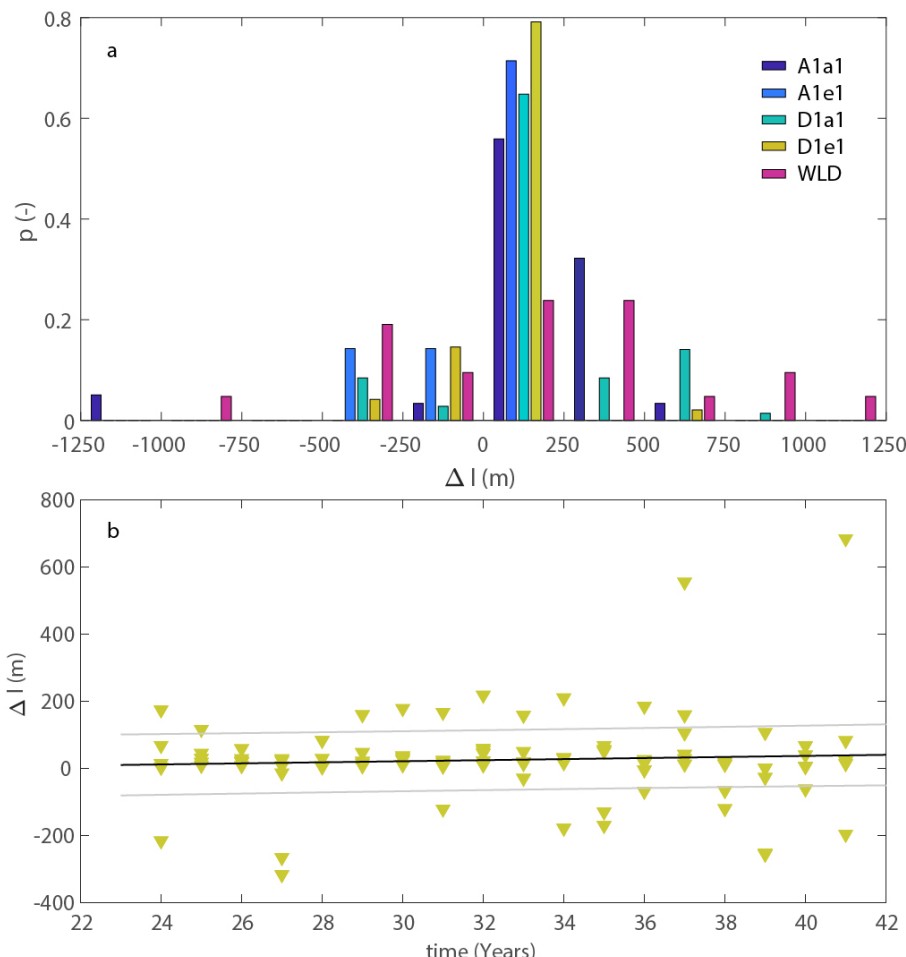

**Figure 6. (a) Histogram of Δl for numerical model A1a1 (dark blue), A1e1 (light blue), D1a1 (seafoam) and D1e1 (yellow) compared with measurements from the Wax Lake Delta (pink). All histograms are binned at 250 m intervals. Descriptive statistics of these populations are shown in Table 2. (b) The location of Δl as a function of time (model years) for delta run 'D1e1.' A linear fit (black line) with 50% confidence interval (gray lines) are shown. The trend of this fit is not statistically significant, and small compared to variation within Δl.**

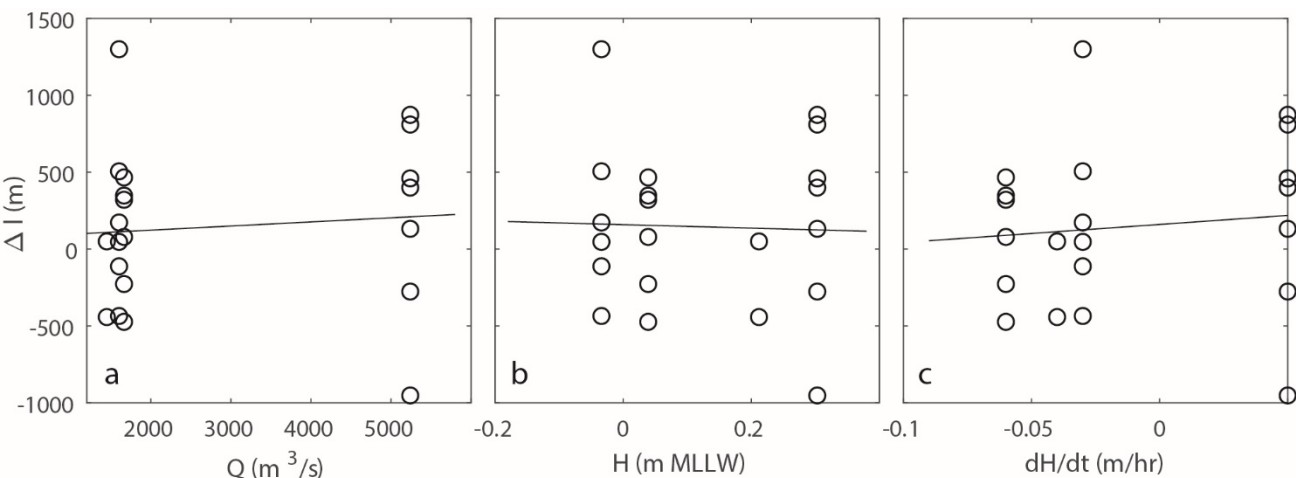

**Figure 7. Comparison of Δl measurements on the WLD to unsteady hydrodynamic conditions. In each case, the black line shows the linear trend, which is negligible. (a) shows upstream water discharge, with $r^2 = 0.01$. (b) shows tidal elevation relative to mean lower low water with $r^2 = 0.001$, (c) shows rate of change of tide elevation averaged over 30 minutes with $r^2 = 0.01$.**

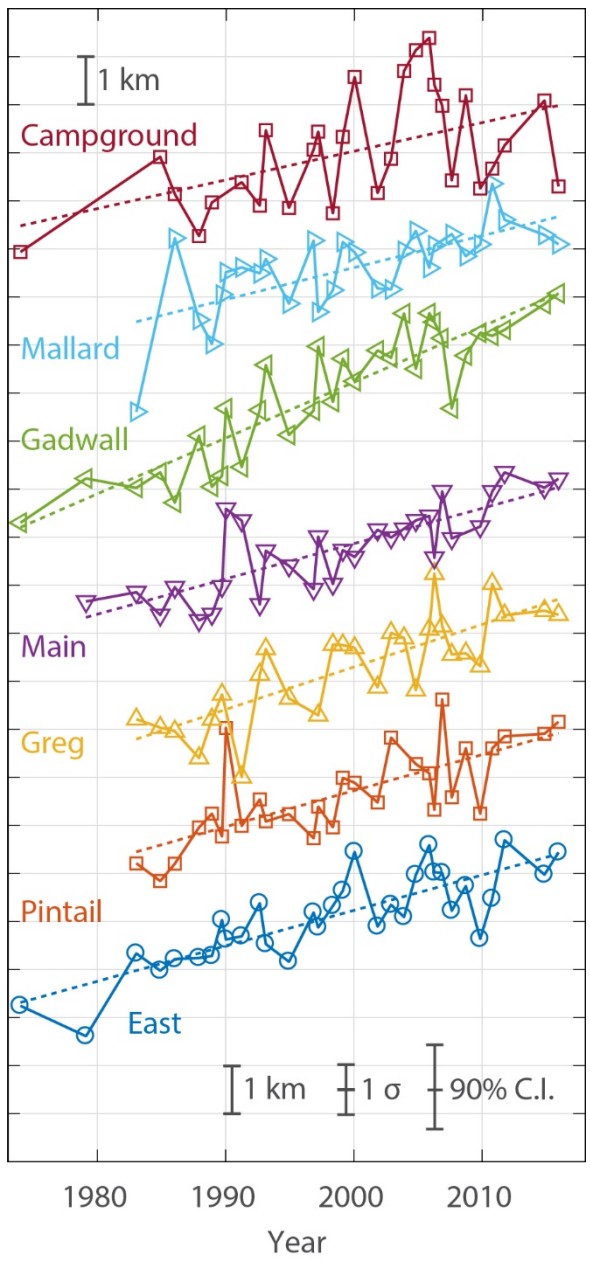

**Figure 8. Growth of individual channels over time. Each series is plotted at the same scale (horizontal lines = 1 km, vertical lines 10 years), but are shifted vertically for clarity. The uncertainty associated with Δl measured at Wax Lake Delta (Figure 6a) is shown with a standard deviation (σ) and 90% confidence interval (C.I.) at the bottom. The primary distributary channels are shown from West to East: Campground Pass (maroon squares), Mallard Pass (turquoise right-pointing triangles), Gadwall Pass (green left-pointing triangles), Main Pass (purple down-pointing triangles), Greg Pass (yellow up-pointing triangles), Pintail Pass (red squares), and East Pass (blue circles). Dashed lines show linear regressions of each dataset.**

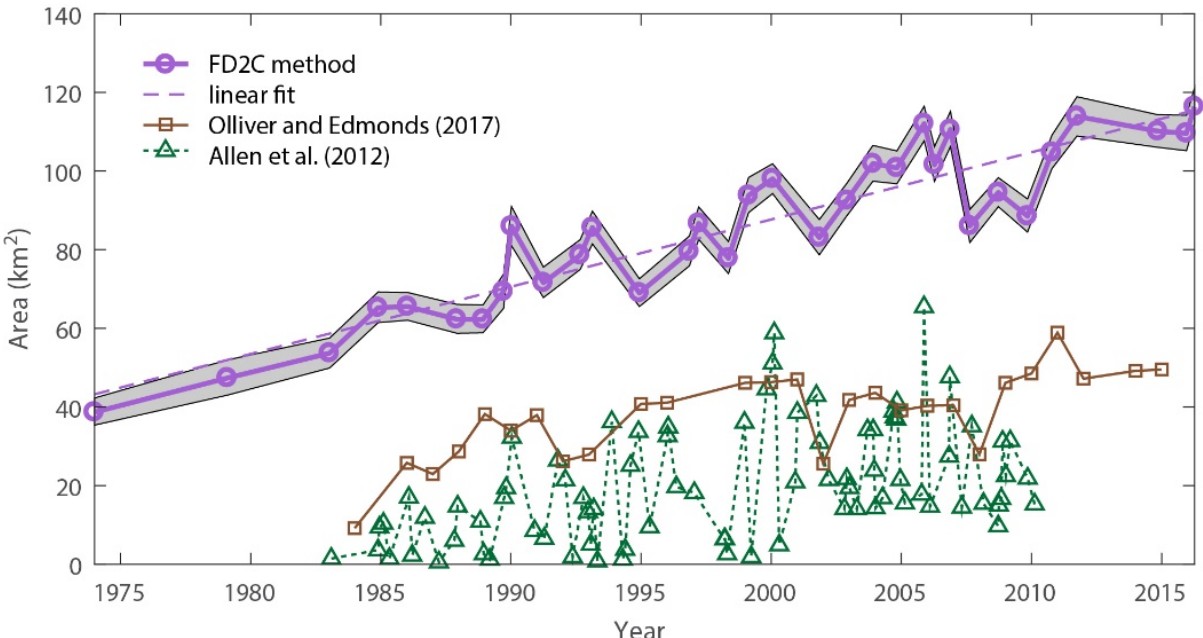

**Figure 9. Area of the Wax Lake Delta as a function of time. Purple circles show the growth of the total delta area of the Wax Lake Delta using the FD2C method. The gray region shows the 1σ deviation (16th to 84th percentile) of area found from Monte-Carlo sampling of Δl (Section 4.1). The dashed line shows the linear fit, with a growth rate of 1.72 ± 0.13 km²/yr. Brown squares and greed triangles show the subaerial area as documented by Olliver and Edmonds (2017) and Allen et al. (2012) over time.**

|          | WLD  | A1a1  | A1e1 | D1a1 | D1e1 |
|----------|------|-------|------|------|------|
| Mean     | 145  | 140   | -44  | 142  | 185  |
| Median   | 132  | 196   | 12   | 52   | 76   |
| Std      | 522  | 434   | 182  | 273  | 284  |
| Iqr      | 701  | 233   | 156  | 171  | 254  |
| Skew     | 0.09 | -3.59 | -1.45| 0.77 | 1.98 |
| Min      | -952 | -1849 | -499 | -398 | -316 |
| Max      | 1300 | 521   | 156  | 965  | 1794 |
| n        | 21   | 79    | 14   | 98   | 185  |
| n misses | 0    | 20    | 0    | 27   | 0    |

**Table 1. Statistic describing the distribution of Δl for the Wax Lake Delta (WLD) and four delta simulations. Std = standard deviation, Iqr= interquartile range, Min and Max = minimum and maximum, n = number of measurements, n misses = the number of measurements where Δl could not be computed.**