# Peer review of "Measuring Subaqueous Progradation of the Wax Lake Delta with a Model of Flow Direction Divergence"

_Earth Surface Dynamics, 2018_

## Referee Comment (RC1) · Anonymous Referee #1 · 19 Jul 2018

Shaw et al. submitted a study titled "Measuring Subaqueous Progradation of the Wax Lake Delta with a Model of Flow Direction Divergence", focused on the use of streaklines from aerial imagery to derive channel tip locations of the Wax Lake Delta. Their main conclusion is that flow convergence occurs at a distance ∼130m downstream of the channel tip.

General comments The study is reasonably well written and organized. My major concern is one of the study's significance. The authors have devised a method to find flow divergence and convergence. The distance between flow divergence and the channel tip (growth rate) is brought as the main contribution of this study. I personally

do not understand the significance. Is there anything here we can learn about process? It is good to see a comparison with Delft3D, but again the authors do not interpret their results. Why is flow diverging/converging? What are the morphodynamics that lead to this behavior? Do the authors expect the same behavior for other deltas? Why (not)?

A second, derived, conclusion of this study is about delta aerial growth, which the authors extract from channel tip locations. In this section there is also no interpretation or discussion about process understanding that can be derived from this data. Does this view of delta area change we way we think of delta morphodynamics, in general or specifically of the Wax Lake Delta?

Overall, I have to conclude that the study does not address a relevant scientific question, and that a shift towards process understanding would require a significant departure from the presented manuscript.

Specific comments P2/3: Section 2 reads like an unorganized mix of different topics ranging from river mouth bars to flow patterns to hydrological connectivity and streak lines. I would ask for better organization and preferably subheadings.

P3L7: remove "strong". Both Leonardi and Nardin modeled relatively low energy marine environments.

P3L18: I strongly suspect streaklines do not track depth-averaged flow, but rather that this case study was performed in a setting where surface flow directions are a good approximation of the depth-averaged flow.

P3L19: how can Shaw et al (2016b) claim reasonable accuracy if validation was done months after the remote sensing images were obtained. I would rephrase this to read more like: "despite limitations in the validation, Shaw et al found reasonable agreement between streaklines and morphology. . ." or similar.

P5L11: what is Dcr?

Fig 1: difficult to read. Perhaps here or in figure 2 explain the structure of the divergent/convergent streak lines.

P5L26: what is a "7% uncertainty for a delta"?

P6L6: the median delta-l for the modelled deltas are within the range of the grid size of the model. Is delta-l even significantly different from a zero mean?

P8L23: with steady boundary conditions Delft3D produced a "significant distribution in delta-l" so winds/tides are unlikely to be a major concern. The authors then follow with a statement that Delft3D variability was less than half the Wax Lake delta variability. So winds/tides could a significant factor?

P10: Why is this a better characterization of delta growth? There are still deltaic deposits beyond the channel tips.

---

## Short Comment (SC1) · 6 Aug 2018

We thank the reviewer for their comments. We agree with many of them: Nardin and Leonardi do indeed model marine environments with microtidal regimes. The D_cr variable (P5L11) is left over from a previous nomenclature and should be removed. The 7% uncertainty was referring to the error (701 m) relative to the delta length (10 km). Further, we agree that the mean $\Delta l$ in some cases is indistinguishable from zero in some cases, and that the treatment of sediments beyond channel tips could be improved. For nearly all of the short comments, we see small changes that can improve our manuscript. We will respond to each of these comments directly at the

end of the discussion period.

However, we disagree with the reviewer's assertion that our paper lacks significance or lacks advances to process-based understanding. We would argue that the processes investigated in our study are (first) fluid flow over a complex, self-formed channel tip, and (second) delta growth. These are both earth surface processes that fall within our reading of this journal's scope. The expansion and contraction of fluid flow over a self-formed distributary channel tip is certainly a processes to us. The reviewer might be seeking a dynamic understanding of the conditions that cause the process to occur, and we cannot provide that yet. Fluid flow on a low-Froude number delta front is influenced by non-local aspects of the bed and flow field, making a simple scaling difficult. Even so, we review the existing literature about flow contraction (Page 4 lines 6-14). We hope that the discovery and validation of the process that we document will pave the way for a detailed dynamic understanding of flow expansion and contraction over complex surfaces in the future.

The second process is delta growth. Our study allows the extension of subaqueous channel tips and subaqueous delta area to be characterized for the first time using remote sensing (one of this journal's objectives). We do not claim that this method is a "better characterization of delta growth." Instead, we argue that delta growth is a complex process, and multiple approaches can lead to an understanding of this complexity. We make several conclusions about delta growth on the Wax Lake Delta from this data (Section 4.2), and relate it to existing theory such as soft avulsion between channels and radially symmetric growth of deltas (Section 5.2). We also looked carefully for breaks in growth rate in the data that would indicate a possible process change, but could not find any unequivocal shifts or trends beyond linear. Linear growth rates are not a jaw-dropping finding, but they do contrast with interpreted breaks in growth sub-aerial rate found by both Allen et al. (2012) and Olliver and Edmonds (2017) (See P10L1-10). Linear growth rates are also valuable validation for certain models of delta growth that require many simplifying assumptions (Kim et al., 2009). Hence, our find-

ings help us better understand the Wax Lake Delta, and provide a case to test on other similar delta fronts such as those shown in Figure 1. In our initial submission, it seemed better to focus primarily on the method, but we will consider adding more detail and context to the progradation rates upon revision.

Allen, Y. C., Couvillion, B. R. and Barras, J. A.: Using Multitemporal Remote Sensing Imagery and Inundation Measures to Improve Land Change Estimates in Coastal Wetlands, Estuaries Coasts, 35(1), 190–200, doi:10.1007/s12237-011-9437-z, 2012.

Kim, W., Mohrig, D., Twilley, R., Paola, C. and Parker, G.: Is it feasible to build new land in the Mississippi River delta, EOS Am. Geophys. Union Trans., 90(42), 373–374, 2009.

Olliver, E. A. and Edmonds, D. A.: Defining the ecogeomorphic succession of land building for freshwater, intertidal wetlands in Wax Lake Delta, Louisiana, Estuar. Coast. Shelf Sci., doi:10.1016/j.ecss.2017.06.009, 2017.

---

## Referee Comment (RC2) · Anonymous Referee #2 · 12 Aug 2018

This article presents and validates a technique for estimating the location of subaqueous channel tips based on the divergence of the flow field on the foreset of river deltas. The technique is referred to as the "Flow Divergence to Channel Tip" (FD2C) model. It builds on the previous work of Shaw et al(2016, JGR-ES) in which the authors first presented the method of mapping "streak lines" to estimate flow direction offshore of the Wax Lake Delta. Here the use of streak lines is used to provide input data (divergence field) for the FD2C model.

The authors claim that flow patterns near subaqueous channel tips follow a specific pattern: that horizontal flow vectors are away from the channel in between the shoreline and the channel tip, where flow is lost to the overbank (positive divergence); and that horizontal flow vectors are towards the water's flow path downstream of the tip (negative divergence). The channel tip is therefore indicated by regions of zero flow divergence.

If the proposed model of flow pattern is widespread, and if streaklines are faithful indicators of flow direction, then this method is a remote sensing technique that can be used to obtain bathymetric information on the foresets of prograding deltas. This region is quite turbid in most deltas, so remote sensing of bathymetry is usually difficult or impossible. The streakline technique combined with the FD2C framework could be useful if the proper conditions are met.

The article is well written, well sourced, and the mathematics are clearly presented. There are valid questions about the universality of the technique, including:

Steaklines might not be good flow indicators everywhere, and subject to wind and tide forcing.

Flow convergance offshore of channel tips may not be universal.

Applying the model requires making some measurements or assumptions to justify the choice of delta-l.

However, the authors mostly address these limitations head on, and provide potential users of the method with the tools to decide whether it might be applicable in their own setting. Given the clarity of the presentation here, other scientists should find it straightforward to apply this technique to their own work. Whether those studies will confirm that the assumptions are valid across many locales remains to be seen, but I expect this paper to be read and the technique to be used by other workers.

I support publication in ESurf with only minor revisions, listed below.

P5L11: Dhat_cr is location where Dhat is zero? unclear

P3L27: It would be good to specify that these are spatial accelerations, to avoid confusion

P6L11: Here you fit a regression line to time vs. delta-l. The slope was small, but the t-test showed that you couldn't reject the null hypothesis of no trend (i.e. zero slope). So doesn't that mean that there might indeed be a trend, and therefore that you cannot say for sure that stationarity exists? My suggestion would be to show the regression line in figure 6 along with error bounds. That should be pretty clear that whatever trend exists is small, and confirm the visualization.

P7L10: I don't see what distribution on delta-l is being assumed for the Monte Carlo simulation. Is it simply uniform over the grey boxes in Figure 5?

Figure 3: If I'm understanding this correctly, the method shown is to estimate the paths of the channels, then extend the channel line beyond the last known channel tip location, then calculate divergence based on streak lines, then use the divergence field to locate the channel tips. So the method shows the distance that the channel tip is along a known or assumed flow path, but doesn't necessarily identify the lateral location of the tip. That means that some information about the channel's path in the subaqueous reach beyond the shoreline is necessary. I think that should be mentioned in the text.

---

## Author Comment (AC1) · 25 Sep 2018

We thank the two reviewers for taking the time with our work, and providing valuable comments which we have addressed, improving the manuscript in the process. Below, we lay out our responses to their comments. We put reviewer comments in brackets. However, it is worth noting at the outset that their comments have led to two new components of the paper. First, in section 3.3 and figure 7, we have added an analysis of the distance between channel tip and critical divergence point (delta-l) as a function of unsteady hydrodynamic conditions. Second, in section 5.2.2 we have further interpreted the linear subaqueous landbuilding growth rates in comparison to the re-

duction in growth rate observed with subaerial monitoring techniques, and interpreted the findings as a result of increased island width, rather than any reduction in sediment accumulation.

We hope that the reviewers and AE find our new manuscript compelling and worthy of publication in ESurf.

[Anonymous Referee #1

General comments The study is reasonably well written and organized. My major concern is one of the study's significance. The authors have devised a method to find flow divergence and convergence. The distance between flow divergence and the channel tip (growth rate) is brought as the main contribution of this study. I personally do not understand the significance.]

The improved understanding of the delta front flow field presented here is significant because it provides a tool for measuring changes to subaqueous channel tips of the Wax Lake Delta using remote sensing for the first time. As described in the introduction, we currently have no way of measuring rapid and important subaqueous changes on deltas where expensive and time-consuming bathymetric surveys are not an option. Demonstrating a method that harnesses understanding of delta front processes represents progress toward monitoring the subaqueous realm, even if it is for only one delta.

[Is there anything here we can learn about process? It is good to see a comparison with Delft3D, but again the authors do not interpret their results. Why is flow diverging/converging? What are the morphodynamics that lead to this behavior? Do the authors expect the same behavior for other deltas? Why (not)? ]

While our study's main contribution is the description of this flow pattern within remote sensing imagery. We also address the "why" question. The transition from flow direction divergence to flow direction convergence ãĂŰ(xãĂŮ_D ÌŇ ) is associated with the

bed transitioning from an adverse bed slope to a basinward one at the channel tip. We have expanded the introduction to this process in the introduction by including a qualitative explanation for what is happening (P4L9). In our opinion, the referenced mechanistic explanations of diverging converging flow are already compelling and capture the process reasonably well. Our contribution is to establish that it can be observed via remote sensing and used for delta monitoring. We have also expanded the discussion of what is required for the method to be applied in other systems (P9L23).

[A second, derived, conclusion of this study is about delta aerial growth, which the authors extract from channel tip locations. In this section there is also no interpretation or discussion about process understanding that can be derived from this data. Does this view of delta area change we way we think of delta morphodynamics, in general or specifically of the Wax Lake Delta?]

We interpreted the data based upon recent theory and measurements from the Wax Lake Delta and other prograding deltas in Section 5.2. The individual channel progradation rates were discussed for the single channel with significantly larger progradation rate and the apparent lack of soft avulsion within the data Section 5.2.1). In the new manuscript we have significantly expanded the discussion of area growth because it is interesting that subaerial growth rates have reduced while the subaqueous ones have remained constant. This is likely due to the importance of channel proximity for marsh aggradation, and has important implications for the Wax Lake Delta and coastal restoration initiatives (Section 5.2.2). We thank the reviewer for prodding us on this, because our interpretation of decoupling or progradation and aggradation is novel and could be a very important process on Wax Lake Delta and controlled sediment diversions elsewhere.

[Overall, I have to conclude that the study does not address a relevant scientific question, and that a shift towards process understanding would require a significant departure from the presented manuscript.]

We disagree with this assessment. The relevant scientific frontiers addressed here are (a) fluid-morphology interaction on a complex delta front and (b) delta evolution. The contribution to a process understanding of frontier (a) is the discovery of widespread convergence of flow direction near channel tips that is consistent and predictable with quantitative uncertainty. The contribution to frontier (b) is the application of this pattern to the first remote-sensing analysis of the subaqueous portion of a prograding delta, revealing remarkable similarities and differences compared to subaerial growth rates.

[Specific comments P2/3: Section 2 reads like an unorganized mix of different topics ranging from river mouth bars to flow patterns to hydrological connectivity and streak lines. I would ask for better organization and preferably subheadings.]

We agree that the paragraph on P2/3 was too long and have broken it up into three components. Section 2 now has an introductory paragraph, a paragraph on delta front bed morphology, a paragraph on flow patterns and streaklines, and a paragraph on flow direction convergence. We hope that this organizes the section.

[P3L7: remove "strong". Both Leonardi and Nardin modeled relatively low energy marine environments.] Removed.

[P3L18: I strongly suspect streaklines do not track depth-averaged flow, but rather that this case study was performed in a setting where surface flow directions are a good approximation of the depth-averaged flow.]

We agree that three-dimensional flows can make it so streaklines do not track flow direction. We also agree that in the case of the Wax Lake Delta, this does not appear important. We now add that "streaklines should indicate flow direction where three-dimensional flow patterns and unsteady changes to flow are minimal (P3L19)."

[P3L19: how can Shaw et al (2016b) claim reasonable accuracy if validation was done months after the remote sensing images were obtained. I would rephrase this to read more like: "despite limitations in the validation, Shaw et al found reasonable agreement

between streaklines and morphology: : :" or similar. ]

We have changed the text to read: "Shaw et al. (2016b) showed that such streaklines depict similar flow directions to direct measurements months apart. (P3L17)"

[P5L11: what is Dcr?]

Dhat_cr was an incorrect remnant from a previous notation scheme. The correct term was $x\_D\grave{\text{I}}\check{\text{N}}$, which is the critical divergence point along an axial channel transect. This line has been amended (P5L12)

[Fig 1: difficult to read. Perhaps here or in figure 2 explain the structure of the divergent/convergent streak lines.]

We have altered Figure 2 to label the diverging and converging flow direction zones.

[P5L26: what is a "7% uncertainty for a delta"?] We find that the sentence is needless and somewhat confusing so we have deleted it.

[P6L6: the median delta-l for the modelled deltas are within the range of the grid size of the model. Is delta-l even significantly different from a zero mean?]

We agree that delta-l is in some cases (A1e1) indistinguishable from zero. However, in other cases it is definitely not equal to zero. Hence, we only choose to conclude that delta l is generally positive and on the order of a few hundred meters or less.

If this comment was asking how a median value could be smaller than the grid resolution, this is because $x\_D\grave{\text{I}}\check{\text{N}}$ is linearly interpolated from $D\grave{\text{I}}\check{\text{N}}(x)$, it's location is given at a subgrid scale. We write this in lines P5L18.

[P8L23: with steady boundary conditions Delft3D produced a "significant distribution in delta-l" so winds/tides are unlikely to be a major concern. The authors then follow with a statement that Delft3D variability was less than half the Wax Lake delta variability. So winds/tides could a significant factor?]

We have addressed this issue with a new figure showing the effect of discharge and tides on measured values of delta l from the Wax Lake Delta (Figure 7). As discussed on P6L3-10, these analyses show that there is not a significant effect of these parameters on delta l. Unfortunately, there was insufficient data to assess the effect of wind setup on delta l (P8L27), although we argue that it is unlikely to have a major effect given the wide distribution of delta l even in a single image.

[P10: Why is this a better characterization of delta growth? There are still deltaic deposits beyond the channel tips. ]

We do not claim that this method is a "better characterization of delta growth." Instead, we argue that delta growth is a complex process, and multiple approaches can lead to an understanding of this complexity. This is exemplified in our new interpretation of a gradual decoupling of progradation and aggradation rates (P10L25)

[Anonymous Referee #2

There are valid questions about the universality of the technique, including: Streaklines might not be good flow indicators everywhere, and subject to wind and tide forcing. Flow convergence offshore of channel tips may not be universal. Applying the model requires making some measurements or assumptions to justify the choice of delta-l.

However, the authors mostly address these limitations head on, and provide potential users of the method with the tools to decide whether it might be applicable in their own setting. Given the clarity of the presentation here, other scientists should find it straightforward to apply this technique to their own work. Whether those studies will confirm that the assumptions are valid across many locales remains to be seen, but I expect this paper to be read and the technique to be used by other workers.]

We appreciate that our effort to address the limitations of the model went over well. We have tweaked the discussion section addressing limitations to say "If the model can be validated in similar settings," which makes it clear that we have no evidence that it will

work beyond the Wax Lake Delta and the models, yet.

[I support publication in ESurf with only minor revisions, listed below.]

[P5L11: Dhat_cr is location where Dhat is zero? unclear]

Dhat_cr was an incorrect remnant from a previous notation scheme. The correct term was x_D ÌŇ , which is the critical divergence point along an axial channel transect. This line has been amended (P5L15)

[P3L27: It would be good to specify that these are spatial accelerations, to avoid confusion] We agree that this needs to change. We have changed "acceleration" to "spatial velocity change." As discussed in Shaw et al. 2016, A_check does not have units of velocity. The change can be found on P3L27.

[P6L11: Here you fit a regression line to time vs. delta-l. The slope was small, but the t-test showed that you couldn't reject the null hypothesis of no trend (i.e. zero slope). So doesn't that mean that there might indeed be a trend, and therefore that you cannot say for sure that stationarity exists? My suggestion would be to show the regression line in figure 6 along with error bounds. That should be pretty clear that whatever trend exists is small, and confirm the visualization.]

We like this approach because it doesn't rely too much on a "failure to reject." We have changed figure 6 to add a linear trend and a 50% confidence interval. We also updated the text to read "The slope that was found (1.6 $\pm$ 2.6 m/yr for D1e1) would introduce a small error to $\Delta l$ relative to the uncertainty of $\Delta l$ (order 100 m) even if it slowly grew over many decades. This near-stationarity suggests that $\Delta l$ can be assumed constant in time, even as a delta progrades. (P6L27)"

[P7L10: I don't see what distribution on delta-l is being assumed for the Monte Carlo simulation. Is it simply uniform over the grey boxes in Figure 5?]

The Monte Carlo sampling was performed by randomly sampling one of the 21 values of $\Delta l$ that was measured at Wax Lake Delta. Hence, no distribution was assumed.
Perhaps with confusion stemmed from referencing Fig. 5a (incorrect) rather than Fig. 6a (intended). The text now reads: "The location of x_$\eta$ İĆ was determined by randomly sampling (with replacement) one of the 21 measured values of $\Delta$l that were measured on the Wax Lake Delta (Figure 6a), and then estimating the location of the channel tip (x_$\eta$ İĆ =x_D İŇ -$\Delta$l)." (P7L23)

[Figure 3: If I'm understanding this correctly, the method shown is to estimate the paths of the channels, then extend the channel line beyond the last known channel tip location, then calculate divergence based on streak lines, then use the divergence field to locate the channel tips. So the method shows the distance that the channel tip is along a known or assumed flow path, but doesn't necessarily identify the lateral location of the tip. That means that some information about the channel's path in the subaqueous reach beyond the shoreline is necessary. I think that should be mentioned in the text.]

The reviewer understands the method correctly. We mentioned this briefly in Section 3.2, but have now expanded the section to be explicit. It now reads: "The method is designed to estimate one channel tip location that is along the subaqueously defined distributary channel axis. The benefit is that this channel axis is easily defined in imagery, but it means that the method cannot account for bends or branches in the subaqueous reach. (P5L21-23)"

---

## Author Response (AR2)

We thank AE Wiberg for her consideration of our resubmitted paper, and we find her few comments to be improvements in every case. We have addressed them in our new manuscript. In addition to addressing her comments, we have made a variety of other minor edits. The content in completely unchanged, but sentences have been rearranged for clarity and style. The greatest of these changes was to change $x$ and $y$, the coordinate system, to $x'$ and $y'$ so as not to conflict with distance along the transect $x$. This change was primarily to Section 3.1.

Please find the AE's comments below in bold, and our responses in plain text.

**Consistently use subaerial (P1: L25, L26) or sub-aerial (P1: L26).**

We now use subaerial throughout the manuscript

**"Slopes" is used as a noun (P2: L30) and as a verb (P3: L1). Maybe unavoidable but it can be a little confusing.**

We now only use only use slope as a noun for clarity.

**P3: L13: over what reach? The levees? But are they channelized?**

We have amended this text to focus on connectivity across the levees (P13L15 below).

**P3: L21: "deltas with established marshes that flow into freshwater basins" is confusing. What is flowing into freshwater basins?**

**P3: L21: change to "or where river discharge is enough …"**

We now write: "Our cursory analysis suggests that streaklines form mostly on deltas with established marshes that are building into freshwater basins or basins where river discharge is enough to make the proximal receiving basin fresh." (P3L23-25 below)

**P5: L9: May be clearer as "For numerically modelled deltas (Figure 4) …"**

We have changes this text to "For numerical models (Figure 4)…" (P5L13)

**P5: L12-13. This sentence is a little hard to follow. Could it be rephrased as something like: "We test the FD2C model by comparing the relative locations of channel tips and the critical divergence point xD on the Wax Lake Delta and on FD2C model grids, as well as on a set of numerically modelled deltas."?**

The previous wording of that sentence was indeed misleading. We now write, "We test the FD2C model by comparing the location of channel tips to the critical divergence point $x\_D˘$ on the Wax Lake Delta and numerical model, as well as on a set of numerically modelled deltas." (P5L18).

**P5: L26: Maybe better as "testing the FD2C model, which indicates that …"? [A model can't "state" anything.]**

Changed as suggested.

**P6: L8: "200 m variation in …"**

Changed as suggested.

**P8: L14: "This uncertainty is generally …"**

Changed as suggested.

**P9: L9: Do you really mean the slope of the slope?**

This is been changed to refer to the bed slope of channels (P9L18 below)

**P9: L10: "shed more light"?**

5   Changed as suggested.

**P9: L26: "revealed at least 10 deltas globally with detectable streaklines (e.g. Fig. 1)"?**

We now write, "our cursory search has revealed at least 10 deltas globally (e.g. Fig. 1) with detectable streaklines under certain conditions" (P10L1 below).

**P9: L31: Does this specifically refer to Mississippi River delta?**

10   We have tweaked this sentence to show that we refer to the Wax Lake Delta here (P10L5).

**P11: L15: change "Carbon" to "carbon".**

Corrected.

**P11: L21: The use of the term "emergent" here could be confusing as it could refer to behavior or position relative to MSL.**

15   We have changed this sentence to read, "The morphodynamic evolution of channel mouths can produce flow patterns and bed morphology that are closely coupled" P12L2 below.

[revised manuscript text omitted]